# Does Adoption of Honeybee Pollination Promote the Economic Value of Kiwifruit Farmers? Evidence from China

**DOI:** 10.3390/ijerph19148305

**Published:** 2022-07-07

**Authors:** Shemei Zhang, Jiliang Ma, Liu Zhang, Zhanli Sun, Zhijun Zhao, Nawab Khan

**Affiliations:** 1College of Management, Sichuan Agricultural University, Chengdu 611130, China; 14036@sicau.edu.cn (S.Z.); zhangliuleo@sina.com (L.Z.); 2Institute of Agricultural Economics and Development, Chinese Academy of Agricultural Sciences, 12 Zhongguancun South Dajie, Haidian District, Beijing 100081, China; zhaozhijun@caas.cn; 3Institute of Agricultural Development in Transition Economies (IAMO), Theodor-Lieser-Str. 2, 06120 Halle (Saale), Germany; sun@iamo.de

**Keywords:** honeybee pollination, artificial pollination, kiwifruit, resilient livelihoods, bivariate probit model, China

## Abstract

Honeybee pollination plays a significant role in sustaining the balance and biodiversity of sustainable rural development, agricultural production, and environments. However, little research has been carried out on the agricultural and economic benefits of pollination, especially for small farmers. This study investigated the adoption of honeybee pollination and its impact on farmers’ economic value using primary data from 186 kiwifruit farmers in three major producing districts, such as Pujiang, Cangxi, and Dujiangyan, in the Sichuan province of China. This study was conducted in two different steps: first, we used a bivariate probit model to estimate factors influencing honeybee pollination and artificial pollination adoption; second, we further used the Dynamic Research Assessment Management (DREAM) approach to analyze the influence of the adopted honeybee pollination economic impact. The results showed that: (1) growers with higher social capital, proxied by political affiliation, are more aware of quality-oriented products, and older growers tend to choose less labor-intensive pollination technology; (2) with the increase in labor costs, more kiwifruit growers would choose honeybee pollination, and more educated growers, measured by the number of training certificates, are more likely to adopt honeybee pollination; (3) the lack of awareness and access to commercial pollinating swarms hinders the adoption of honeybee pollination; (4) in addition to the economic benefit to producers, honey pollination also brings an even larger consumer surplus. This study suggests some policy recommendations for promoting bee pollination in China: raising farmers’ awareness and understanding of bee pollination through training, promoting supply and demand in the pollination market, and optimizing the external environment through product standardization and certification.

## 1. Introduction

Honeybee pollination plays a significant role in maintaining the balance and biodiversity of sustainable agricultural systems, rural development, and ecosystems [1,2]. Numerous studies have shown that honeybee pollination is critical to the vegetation process of insect crops such as grains, oilseeds, fruits, and pastures [3,4]. Honeybee pollination can increase yield, improve quality [5], and enhance seed vigor, thus becoming an important measure to promote the agricultural sector, reduce product costs, save labor and time, and ensure food safety, quality, employment, production practices, and livelihood [6,7]. With population growth and affluence in many developing countries, world food demand continues to increase [8,9,10]. In addition to the quantity of food, people’s demand for diversified and high-quality agricultural products is also increasing with the improvement of consumption levels [11,12]. The most populous and fastest-growing countries, including China, face the enormous challenge of protecting their deteriorating environment and ecosystems while meeting growing food demand and quality [13,14]. In 2018, the Chinese government decided to change course and launch a high-quality development strategy to meet growing demands without further damaging the environment and the development opportunities of future generations [15]. As shown by this development concept, agricultural production methods are envisaged to be greener and more sustainable, and bee pollination is an important part of achieving sustainable agricultural production [16].

With the increasingly prominent drawbacks of the stallholder farmer income, the Chinese government has introduced a series of policies and measures to promote land transfer [17]. The moderate-scale operation of family farms and farmers’ cooperatives has been gradually promoted [18]. At the same time, the transformation of the agricultural industrial structure to a specialized and regionalized layout has accelerated. The structural shift in Chinese agricultural production to larger-scale and particular operations in certain industries (e.g., fruit production) and regions has increased the demand for bee pollination [4]. The planting area of economic crops such as apples, pears, kiwi, cherries, and strawberries has been greatly expanded [4,19,20]. A common feature of these fruits is that they are all cross-pollinated crops, and their large-scale and centralized production requires the tracking and improvement of bee pollination supply systems [21]. Although the rapid and large-scale development of economic crops in China has created a market for pollination demand, the bee pollination service industry has not been well developed [4]. Compared with developed countries, China’s bee pollination industry lags significantly. In the United States, for example, farms and orchards rent more than a million bee colonies each year for pollination, accounting for a quarter of the total bee colony in the country [22,23]. In Canada and Germany, pollinating colonies on fruit trees alone reach 300,000 per year [24,25]. In the UK, honeybees pollinate strawberries, apples, raspberries, and pears with an economic value of up to GBP 600 million [26].

In contrast, artificial pollination or hand pollination has been widely used because most farmers do not have sufficient knowledge of the combined benefits of honeybee pollination [27,28]. Farmers often mistakenly believe that bee pollination is less reliable and less risky than manual pollination. In addition, the small size of the farms and the low willingness to pollinate also reduce the enthusiasm of beekeepers to provide pollination services. Most professional beekeepers rely primarily on honey production for their income. Therefore, artificial pollination is widely used in the market instead of bee pollination services. Fruit and cash crop growers would rather hire labor for hand pollination than hire bee colonies for pollination. The cultivation of Kiwifruit in China is relatively new; Sichuan is the cradle of Chinese kiwifruit and one of the largest producing provinces. The Longmen Mountains area, including Guangyuan, Cangxi, and Dujiangyan, is an ideal place for kiwifruit growth due to its suitable weather, soil, and terrain [29]. In recent years, the kiwifruit industry has developed rapidly, covering an area of 600,000 mu and producing 230,000 metric tons, ranking second in all provinces in the country. Red-fleshed kiwifruit is a unique variety cultivated in Sichuan, which is quickly welcomed by consumers at home and abroad [30]. In addition to local consumption, 50% of the products are exported to major cities such as Beijing and Shanghai, and 20% are exported to more than 20 countries such as Japan, South Korea, the United States, Germany, and the United Kingdom. Because of the high yield and economic benefits of kiwifruit, the local government decided in 2017 to further expand the planting area to 1 million mu in the near future [31]. Kiwifruit cultivation has become a distinctive industry in the Longmen Mountains; tens of thousands of farmers use kiwifruit cultivation as their main source of income [32].

Preliminary investigations have shown that most of the kiwifruit growers in Sichuan Province have opted for artificial pollination [2]. Every year when the flowers bloom in mid-April, farmers collect the powdered male flowers and use a brush to manually pollinate the female flowers, one by one. However, compared with natural pollination, this artificial pollination method has serious defects and deficiencies. First, given the steady rise in labor prices over the past few years, the cost is very high. According to this survey, the labor cost of artificial pollination is about 230 yuan per mu, which is 9.2% higher than the total cost of bee pollination services. Given the labor shortage and consequent wage hikes, the economic disadvantage will only become worse [33]. Second, the quality of the fruit is affected by this artificial pollination. For instance, evidence shows that artificial pollination has higher rates of deformed fruit, which translates into lower prices and overall income. Third, many elderly people, women, and even children are engaged in these cumbersome and sometimes dangerous practices. This is also undesirable in many senses. Furthermore, bee pollination generates positive environmental externalities in addition to generating additional income for beekeepers. Consequently, bee pollination services are rarely used by kiwifruit growers, which is confusing because bees are much better at pollinating than humans and renting beehives for pollination services is cheaper than hand pollination. How to promote the development of bee pollination services to achieve untapped environmental and socioeconomic benefits, with significant social value in supporting sustainable agricultural production, is an interesting economic research question. Therefore, this paper mainly aims to explore: (1) the reasons behind the low adoption rate of bee pollination in practice, although it is considered to be superior in theory; (2) if bee pollination services are used, what would be the economic benefits; (3) what would be the effective strategies and policy interventions to promote the bee pollination services. These problems need to be solved immediately, which would be crucial to promoting the development of the honeybee pollination industry and improving the competitiveness of China’s fruit industry.

The remainder of this paper proceeds as follows. Section 2 presents the data and methodology. Section 3 reports and discusses the empirical results. Section 4 presents the discussion and the final section is conclusions with policy implications.

## 2. Data and Methodology

### 2.1. Survey Data and Variables

This study mainly relies on the household survey data obtained from a questionnaire survey of kiwifruit growers in Cangxi, Pujiang, and Dujiangyan, three kiwifruit-producing districts in Sichuan Province in August 2016, as shown in the map of the study area. (Figure 1). We trained a survey team of five enumerators and two supervisors. First, we trained participants on the questionnaire and conducted a pre-test survey to identify missing and inappropriate questions and other potential problems in the questionnaire. Then, we modified the questionnaire accordingly and collected a total of 224 samples by the random sampling method. A total of 186 valid samples were included in the study, accounting for 83% of the total number of samples. All questions and variables were designed based on expert knowledge, existing literature, and preliminary local surveys.

In this study, the factors affecting farmers’ adoption of pollination techniques were divided into four groups: farmer characteristics, family characteristics, management characteristics, and other relevant control variables. Farmer characteristics mainly refer to farmers’ age, education level, and political background; family characteristics include the proportion of kiwifruit income and the number of kiwifruit farm workers in the family; management characteristics refer to kiwifruit planting scale, planting experience, product certification, grower certification, and whether growers can buy pollinators. In addition, we selected variables such as cooperative membership, farm-to-county distance, and farmers’ beliefs about which method of honeybee pollination or artificial pollination produces better-quality fruit.

### 2.2. Adoption Model of Pollination Technologies

Numerous studies have been conducted on household technology adoption behaviors and have focused on different adoption behaviors. Based on the number of technologies adopted, these behaviors can be classified as adopting a single technology, adopting two technologies at the same time, or adopting multiple technologies, with no difference in utility. The decision of whether to adopt a single technology usually adopts models such as Probit, Tobit, and Logistic to analyze the factors that affect farmers’ decision-making. The simultaneous adoption of two technologies mainly analyzes the decision-making behavior of farmers in the production process in which two related or inseparable technologies are simultaneously adopted, usually using a partially observed bivariate probit model [34,35,36]. In the case of several agricultural techniques, it is generally assumed that there is no difference in the effectiveness of the techniques for farmers. The counting model is used to analyze the adoption behavior of farmers in this situation, and the specific forms are mostly the Poisson model and the negative binomial model (the difference is whether the adoption behavior conforms to the constraints of equal expected value and variance).

Initial surveys showed that 4.3% of farmers used bee pollination alone and 11.3% used both bee and hand pollination, confirming the low adoption rate of bee pollination, as previously described. Therefore, this study adopted an improved bivariate probability model. The model settings are as follows:(1)$$\left\{ \begin{array}{c} y_{1}=\beta_{1}+\beta_{1}^{\prime }X_{1}+\varepsilon_{1}\\ y_{2}=\beta_{2}+\beta_{2}^{\prime }X_{2}+\varepsilon_{2}\\ E(\varepsilon_{1})=E(\varepsilon_{2})=0\\ \mathrm{var}(\varepsilon_{1})=\mathrm{var}(\varepsilon_{2})=1\\ \mathrm{cov}(\varepsilon_{1},\varepsilon_{2})=\rho \end{array}\right.$$

In Equation (1), $y_{1}$ and $y_{2}$ are potential variables that cannot be observed, so they can be understood as pollination utility and are the outcome variables. If $y_{1}>0$, it shows that the effect of using honeybee pollination is positive, the farmers choose honeybee pollination, and $y_{1}=1$; otherwise, $y_{1}=0$. If *y*_2_>0, it shows that the effect of using artificial pollination is positive, the farmers choose artificial pollination, and $y_{2}=1$; otherwise, $y_{2}=0$. $X_{1}$ and $X_{2}$ are the factors affecting farmers’ choice of pollination, $\beta_{1}$, $\beta_{2}$, $\beta_{1}^{\prime }$, and $\beta_{2}^{\prime }$ are estimated coefficients; $\varepsilon_{1}$ and $\varepsilon_{2}$ are random disturbance terms for the normal distribution of the binary joint; ρ is the correlation coefficient between $\varepsilon_{1}$ and $\varepsilon_{2}$. If ρ=0, then $\varepsilon_{1}$ and $\varepsilon_{2}$ are not correlated; if ρ>0, $y_{1}$ and $y_{2}$ are complementary; if ρ<0, $y_{1}$ and $y_{2}$ are replaceable.

With expected yield maximization and resource constraints, kiwifruit growers are faced with a binomial decision between honeybee pollination and artificial pollination. There are four choices between these two pollination techniques: single honeybee pollination, artificial pollination, both used, and neither. Therefore, there is an intrinsic link between the two technologies (optional or complementary, requiring validation), which makes these two selection behaviors somewhat related. In other words, pollination techniques adopted by kiwifruit growers are a function of substitution effects or complementary effects. Furthermore, a large number of empirical studies have revealed that the characteristics of farmers, management conditions, technical environment, and other factors will also affect farmers’ choices [34,35,37].

In this study, $y_{1}$ and $y_{2}$, respectively, represent the farmers’ choice of honeybee pollination and artificial pollination, and when farmers are using honeybee pollination, then $y_{1}=1$; when farmers are not using honeybee pollination, then $y_{1}=0$; when farmers are using artificial pollination, then $y_{2}=1$; when farmers are not using artificial pollination, then $y_{2}=0$. Therefore, the above four results can be reduced to (1, 1), (1, 0), (0, 1), and (0, 0), respectively. Accordingly, to analyze the influence of various factors on the pollination behavior of farmers, this study established a simultaneous bivariate probit model [38]. This model is an extension of the probit model and applies to the two conditions of the following simultaneous equations: (1) there is a correlation between the random perturbation assumption equations, so it is necessary to estimate equations simultaneously; (2) there are two variables in the model. The model settings were as follows:(2)$$y_{1}{}^{*}=\gamma_{1}y_{2}{}^{*}+\beta_{1}X_{1}+\varepsilon_{1}=1\;(y_{1}{}^{*}\;>\;0)$$
(3)$$y_{2}{}^{*}=\gamma_{2}y_{1}{}^{*}+\beta_{2}X_{2}+\varepsilon_{2}=1\;(y_{2}{}^{*}\;>\;0)$$
where $y_{1}^{*}$ and $y_{2}^{*}$ represent the binary latent variables adopted by the observable honeybee pollination and artificial pollination technologies, respectively; *X*_1_ and *X*_2_ are independent variables of personal characteristics, family characteristics, management characteristics, and the technical environment.

### 2.3. Economic Impacts Model

The crucial objective of estimating the economic effects of honeybee pollination is to further compare and analyze the changes in the benefits of consumers and producers brought about by honeybee pollination and artificial pollination technologies to demonstrate the changes in cost benefits and economic benefits that can be realized by adopting honeybee pollination technology. The economic surplus method was used for analysis in this study. The dynamic research evaluation for management (DREAM) approach, developed by the International Food Policy Research Institute (IFPRI), focuses on the changes in the economic benefits between consumers and producers brought by new technologies. The DREAM method was developed into a mature system, which was used to analyze the economic impact of adopting honeybee pollination technology in this study.

The DREAM model focuses on three economic traits: cost, yield, and quality. Thus, the advantages of honeybee pollination lie in saving costs and improving fruit quality. The technical benefits of quality improvement are not reflected in the production; instead, these are reflected in the sales price. Consequently, the economic benefits of cost savings and quality improvements were calculated differently. Both methods are used in this section. First, the economic impact of honeybee pollination on growers and consumers in terms of cost savings and yield increase was calculated using artificial pollination as a reference technology. Second, the economic impact on growers and consumers of higher prices due to the improved quality of kiwifruit was assessed using honeybee pollination. The DREAM model was used to improve quality by using a moving-demand curve to approximate the effect of quality improvement. Under closed-market conditions, it is assumed that the total economic benefits of honeybee pollination technology are calculated, as shown in Figure 2.

*S*_0_ denotes the supply function of the kiwifruit market before the application of honeybee pollination technology, and *D*_0_ denotes the demand function. The initial prices and quantities are *P*_0_ and *Q*_0_, respectively. The application of honeybee pollination technology has improved the quality of kiwifruit products and has increased the demand for these products. As shown in Figure 2, the demand curve moves upward to *D*_1_, the market equilibrium point moves from a to b, the curve moves by $k_{1}$, the demand increases to *Q*_1_, and the price rises to *P*_1_. The reduction in unit production cost is $k_{2}$ (or the translation of the increase in unit output). According to Figure 2, the supply curve moves down to *S*_1_ in parallel, the market equilibrium point moves from b to c, the demand increases to *Q*_2_, and the price rises to *P*_2_.

It can be inferred from the figure that the consumer surplus is equivalent to the area of quadrilateral *P*_0_*’’ecP*_2_ and that the producer surplus is equal to the area of the *P*_2*cd*_
*P*_0_*’* prime. Figure 2 shows the calculation formula of the economic surplus of consumers and producers in the DREAM model, and it is concluded as:(4)$$\mathrm{Producer}\;\mathrm{surplus}=0.5\left(k_{1}+P_{2}-P_{0}\right)\left(Q_{2}+Q_{0}\right)$$
(5)$$\mathrm{Consumer}\;\mathrm{surplus}=0.5\left(k_{2}+P_{0}-P_{2}\right)\left(Q_{2}+Q_{0}\right)$$
where the moving distance of the demand curve is $k_{1}$; that of the supply curve is $k_{2}$; the price and quantity of the final equilibrium market of the market are *P*_2_ and *Q*_2_, respectively. The calculation formulae for the four parameters are as follows: $k_{1}$ is equivalent to the price difference of kiwifruit when honeybee pollination technology is applied compared with that when control technology (artificial pollination technology) is applied, $k_{1}=\Delta P$; $k_{2}$ represents the changes in the cost and yield of honeybee pollination compared with that of the control technology, resulting in the moving distance of the supply curve. This is expressed by the following formula:(6)$$\left\{ \begin{array}{c} k_{t}^{\prime }=\sum_{k=1}^{t}(\frac{y_{t-k+1}}{\varepsilon }-\frac{c_{t-k+1}}{1+y_{t-k+1}})\Delta A_{k}P_{1}\\ P_{1}^{}=P_{0}+\frac{\eta k_{1}P_{0}}{\varepsilon +\eta }\\ P_{2}^{}=\frac{\gamma -\alpha +\delta *k_{1}-\beta *k_{2}}{\beta +\delta }=P_{0}+\frac{\eta *k_{1}-\varepsilon *k_{2}}{\varepsilon +\eta }\\ Q_{2}=\frac{(1-\varepsilon )Q_{0}+\varepsilon Q_{0}}{P_{0}(P_{2}+k_{2})} \end{array}\right.$$
where *y* is the growth rate of kiwifruit yield per unit product under the condition of honeybee pollination compared with that of the control yield, y=ΔY/Y; *c* is the ratio of cost savings per unit product, $c=\Delta C/{\left(Y\times PP\right)}_{i}$; *A* is the adoption rate of honeybee pollination; ε and η represent the supply elasticity and demand elasticity of the kiwifruit commodity, respectively; *Q*_2_ and *Q*_0_ represent the total output of commodities before and after the final market equilibrium, respectively.

## 3. Results 

### 3.1. Statistical Description

Table 1 shows that only 4.3% of kiwi growers used honeybee pollination, 84.4% used artificial pollination, and the remaining 11.3% used honeybee and artificial pollination simultaneously, indicating that the adoption rate of honeybee pollination was very low in the research area. In terms of the intention to use honeybee pollination, 56.7% of the farmers were reluctant to use honeybee pollination because of the lack of honeybee pollination knowledge, and 78.7% of the farmers said that they were worried that changing the current technology route would cause problems, with the impact of bee pollination likely to worsen. Another 15.6% of the farmers did not adopt honeybee pollination, because of the difficulty in purchasing pollination services. Among all surveyed farmers, 66.2% thought that the quality of manually pollinated kiwifruit was better; 44.2% of farmers identified the lack of stable honeybee pollination service providers as the reason for not using honeybee pollination; 23.1% were not aware of the existence and effects of honeybee pollination services; 25.1% of the farmers said that the weather and florescence inconsistency between male and female flowers affected their choice to adopt honeybee pollination. Only 26.09% of honeybee pollination adopters had received technical guidance on honeybee pollination, and their technical services were mainly from the agricultural management departments or associations (55.56%), followed by local beekeepers (29.63%).

In our sample, the average age of kiwi growers is 51.76, and the growers’ education attainment is between junior high school level and high school level. About 21% of the growers in our samples are communist party members, which usually indicates high social capital. Kiwi planting income accounts for 47.16% of the growers’ total family income. Growers hold 123.38 mu kiwi planting areas with 2.26 labor forces engaging in Kiwi planting on average, and their average planting experiences are 8.72 years. An amount of 72% of the growers are cooperative members. The average distance to the closest county capital is 21.04 km. As for the question of which pollination methods produce better fruits, 63% of growers believe that honeybee pollination results in better-quality kiwi fruits, while 19% believe that artificial pollination results in a better quality.

According to the survey, the average production cost was 3854.68 yuan/mu when honeybee pollination is used, while the cost was 4244.58 yuan/mu when artificial pollination is adopted. The cost of honeybee pollination was 389.90 yuan, which is cheaper than that of artificial pollination, and the revenue of honeybee pollination increased by 34.94%. The lower cost and higher revenue of honeybee pollination were mainly attributed to the decrease in labor costs and pollination inputs; specifically, farmers could save 229.85 yuan of the labor cost and 222.23 yuan of the pollination cost per mu with honeybee pollination. The yield per mu was 1073.43 kg/mu with honeybee pollination, which is 142.95 kg/mu higher than that of artificial pollination (see Table 2 for details).

Kiwifruits that were pollinated by honeybees are regarded by customers to be of better quality than those artificially pollinated; therefore, customers are willing to pay a premium for bee-pollinated products. According to the survey, the price of kiwifruit pollinated by honeybees was 1.0–3.0 yuan/kg higher than that of artificially pollinated kiwifruits. The outputs of Dujiangyan, Pujiang, and Cangxi were 45,000, 64,000, and 150,000 tons, respectively. Due to the lack of relevant research, this study valued the market supply and demand elasticity of kiwifruit by referring to the market supply and demand elasticity of related fruits. According to the literature, we valued the supply elasticity and demand elasticity of kiwifruits to be 0.125 and −0.34, respectively.

The S-curve model, instead of the linear model, was used to estimate the adoption rate of honeybee pollination technology, as many bee farmers are not familiar with the honeybee pollination technology and the relevant market supporting services are not perfect. According to the survey, the adoption rate of honeybee pollination technology was only 4.3% in 2016. The adoption of honeybee pollination technology is affected by many factors such as farmers’ awareness, technical services systems, and consumers’ concerns about both quality and price. In this study, we set two estimation scenarios. Scenario 1 is an optimistic estimation, assuming that the current honeybee pollination technology can overcome the existing difficulties and can be applied in various producing areas, with the maximum adoption rate set as 50%. Scenario 2 is a conservative estimate, assuming a slow adoption of technology promotion, and it is only applied to some farmers. We set its maximum adoption rate at 30%, which is seen as the initial adoption rate. Under both scenarios, it is assumed that the starting year is 2016, and the period required for honeybee pollination to reach its maximum adoption rate is 10 years.

### 3.2. Results of the Pollination Adoption Model

The bivariate probability model was used to estimate the factors that affect honeybee pollination and artificial pollination, and the statistical software Stata13.0 used for data analysis. The results are shown in Table 3. The fitness of the model was tested, and the factors considered in this study proved to be robust through various robustness tests. As mentioned in Section 2.2., the relationship between artificial pollination and honeybee pollination is complementary or substitution; when ρ is negative and is significant at the 1% level, it indicates that artificial pollination and honeybee pollination showed a substitution effect, rather than a complementary effect during growth. Farmers who use artificial pollination do not like to use honeybee pollination as an alternative or substitute and vice versa.

(1) Among the variables of farmer characteristics, political affiliation had a positive effect on adopting honeybee pollination (at 10% significance level), suggesting that growers with a significant political orientation are more aware of quality-oriented products. Age had a significant impact on the adoption of honeybee pollination (at 5% significance level), indicating that growers tend to choose less labor-intensive pollination technology with increasing age; however, age and education had less significant effects on artificial pollination.

(2) Among the variables of family characteristics, we found that honeybee pollination had a reverse relationship with labor inputs (at 5% significance level), and labor reduction by 1% could increase the adoption rate by 0.366%, indicating that with the increase in labor costs, kiwifruit growers would seek to adopt honeybee pollination. In terms of the proportion of kiwifruit income, income had no significant effect on honeybee pollination and artificial pollination.

(3) Among the variables of management characteristics, the results showed that planting experiences had significant negative impacts on the adoption of honeybee pollination (at 1% significance level), while it had positive impacts on artificial pollination; one possible reason for this is that the longer the year of planting, the less willing farmers are to alter existing technology routes to embrace new ones. Furthermore, we found that growers with a higher number of certificates (trained farmers) tended to adopt honeybee pollination (at 10% significance level), which means that the more certificates farmers had, the more training they had, and the greater their awareness of the quality and safety of their produce, the higher their likelihood of adopting honeybee pollination. In addition, the results showed that there was a significant positive correlation (at 1% significance level) between farmers’ access to purchase pollinating swarms and the adoption of honeybee pollination. We also found that joining cooperatives did not promote the adoption of honeybee pollination, illustrating that cooperation is not an effective organization to promote growers’ new technology adoption in this sector of China.

(4) We also found that growers had different beliefs about kiwifruit quality when adopting the two pollination technologies in our investigation, and the results reflected that growers tended to choose the technology that they believed to be of better quality (at 5% significance level for honeybee pollination and 10% for artificial pollination). Distance significantly impacted honeybee pollination (at a 5% significance level), which reflects the fact that growers could receive relatively new technology diffusion if they were close to the county; therefore, honeybee pollination is more likely to be adopted.

### 3.3. Results of the Dynamic Research Evaluation for the Management Model

Based on the theory of consumers’ economic surplus and producers’ economic surplus, we estimated the economic surplus of honeybee pollination adoption between 2016 and 2026. The output and price of the kiwifruit market in the first year of application are represented by *Q*_0_ and *P*_0_, respectively (Figure 2). According to our survey, *k*_1_ (price difference) is 1.86 yuan/kg; hence, *Q*_0_ and *P*_0_ are 11.48 million kg and 10.98 yuan/ton, respectively. The price elasticity of supply and demand are 0.125 and −0.34, respectively. The year 2016 was taken as the base year to calculate the economic benefits of bee pollination under the following two scenarios, as shown in Table 4.

As shown in Table 4, when the adoption rate of honeybee pollination reaches 50% under the first scenario, it will bring an economic surplus of 266 million yuan to the producers in the first year. By 2026, the total surplus will reach 2.58 trillion yuan. Consumers will gain more benefits from the adoption of honeybee pollination. In 2016, the economic surplus of consumers reached 393 million yuan, and by 2026, the total consumption surplus will reach 4.888 billion yuan. In 10 years, the total revenue will be 7.468 billion yuan, and the consumer surplus will be 1.9 times that of the producer surplus, implying that when honeybee pollination is promoted, the benefits to consumers are greater than those to producers.

In the second scenario, we set the honeybee pollination adoption rate to 30% in 2026; consequently, the producer surplus will be 257 million yuan, and the consumer surplus will be 408 million yuan. Similarly, consumers will obtain more welfare from honeybee pollination. From 2016 to 2026, the cumulative economic surplus will reach 4.613 billion yuan, compared to 2.77 billion yuan for producers. The consumer surplus will be 1.7 times that of the producer surplus.

## 4. Discussion

The adoption rate of honeybee pollination services in kiwifruit is extremely low, which is consistent with the existing literature and existing preliminary research. This means a loss of environmental and socioeconomic welfare gains. Although numerous studies have verified that bee pollination is cheaper, product quality is better, yields are higher, and income is higher, Sichuan’s fruit farmers are not aware of this common sense. False beliefs persist among farmers and a lack of awareness and knowledge about pollination hinders the adoption of bee pollination services. Low levels of education and a lack of frequent training are to blame for low adoption rates, as evidenced by empirical estimates. In addition, farmers in distant counties are less likely to adopt bee pollination, which may also be related to low access to knowledge and training. While it is not surprising that education and training are positively associated with bee pollination adoption, party members were more likely to adopt bee pollination services. This is about social capital and power, which allows them to gain more knowledge and training, as well as better information on pollination services. In addition, if the government were to promote bee pollination services, party members tended to abide by the policy, which could have a word-of-mouth effect and positive social impact among fruit farmers.

Observed models also indicate that older farmers and households with less labor are more likely to employ bee pollination services. With the general trend of aging in Chinese society, especially in rural areas, and the steady increase in wages due to labor shortages, research suggests that the adoption rate of pollination services should increase in the future. However, this is not inevitable, as farmers tend to be risk-averse, practice what works for them now, and are reluctant to make changes if transaction costs (such as information searches for bee pollination services) are too high. As we detail in the next section, the government needs to facilitate the development of the market for bee pollination services. The DREAM model shows that bee pollination services deliver large untapped welfare gains. Adoption of bee pollination services can generate high economic surpluses for both producers and consumers, even at 30% adoption. It is worth pointing out that the consumer surplus gained from adoption is higher than that of producers. This is encouraging and can serve as an impetus for governments to step in and drive the development of bee pollination services, as society as a whole stands to benefit. This study investigated only one crop in one province, kiwifruit in Sichuan. However, the research yielded that the methods used in this study could provide valuable insights into other crops and regions.

## 5. Conclusions and Implications

### 5.1. Conclusions

The current study investigates the adoption of honeybee pollination and its impact on farmers’ economic level using primary data from 186 kiwifruit farmers in three districts of Sichuan province, China. This study first used a bivariate probit model to estimate the influencing factors of bee pollination and artificial pollination, and then the DREAM method was used to analyze the economic impact of adopting bee pollination. The main findings can be summarized as follows: First, growers with high social capital signified by party members have a higher awareness of quality products, and older growers are more likely to choose bee pollination services with less labor intensity. Second, as labor costs increase, kiwifruit growers will seek to adopt bee pollination, with planting experience having a significant positive impact on the adoption of bee pollination and a negative impact on manual pollination, with more certified growers (trained farmers) tending to adopt honeybee pollination, which means that the more training farmers received, the higher their likelihood of adopting honeybee pollination. Third, there was a significant positive correlation between farmer acquisition of pollinator colonies and bee pollination adoption. However, joining a cooperative did not promote the adoption of bee pollination, consequences that suggest that cooperatives were not an effective organization for promoting the adoption of new technologies. Fourth, growers tend to choose technologies they trust are of better quality; if growers are close to county towns, they have access to relatively new technologies to spread; hence, bee pollination is more likely to be adopted; finally, with the benefits of adopting bee pollination, the existing study found that the benefits of promoting bee pollination were greater for consumers than for producers in two cases (50% and 30% of bee pollination adoption rates, respectively).

### 5.2. Policy Implications

Based on the findings, this study makes some policy recommendations to further promote the adoption of bee pollination and develop the pollination services market. First, support farmers’ understanding and mastery of pollination technology. The beekeeping management department and the national beekeeping technology system should pay attention to the demonstration role in the promotion, lectures, and training of bee pollination technology, and display the main points and effects to farmers through regional comparative experiments. At the same time, it is necessary to carry forward the concept of “respecting production laws and providing high-quality agricultural products”, so that the economic, social, and ecological benefits of honeybee pollination will be widely recognized by farmers. Secondly, establish a complete supporting system to promote honeybee pollination. It is essential to promote the establishment of professional service institutions for bee pollination in fruit areas as soon as possible, cultivate a group of professional talents, and provide guidance and services in terms of bee species selection and layout. For management and recycling, build a bee pollination supply and demand information service platform so that users can receive pollination services and consultations conveniently, quickly, and promptly. Thirdly, optimize the external environment and promote honeybee pollination, provide assistance with the large-scale management and standardization of fruit tree planting, and reduce the cost of honeybee pollination. In addition, strengthen the education and training of farmers and improve the level of production and operation. The high-quality agricultural system also needs to be incorporated into the horizontal structural reform agenda of agricultural supply. In order to improve the market competitiveness of bee-pollinated agricultural products, promote the wide application of bee-pollinated technology and improve the quality with high profit.

## Figures and Tables

**Figure 1 ijerph-19-08305-f001:**
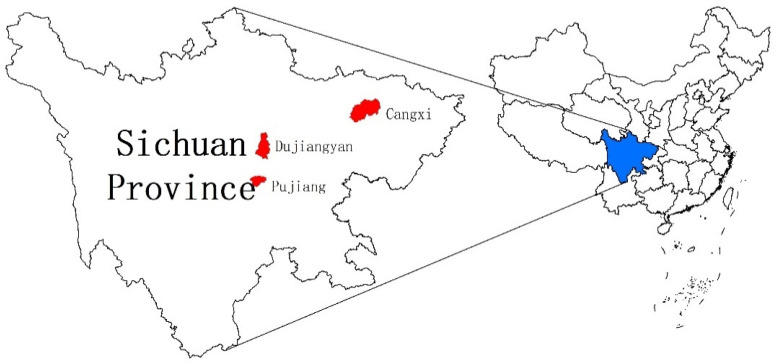
Map of the study area.

**Figure 2 ijerph-19-08305-f002:**
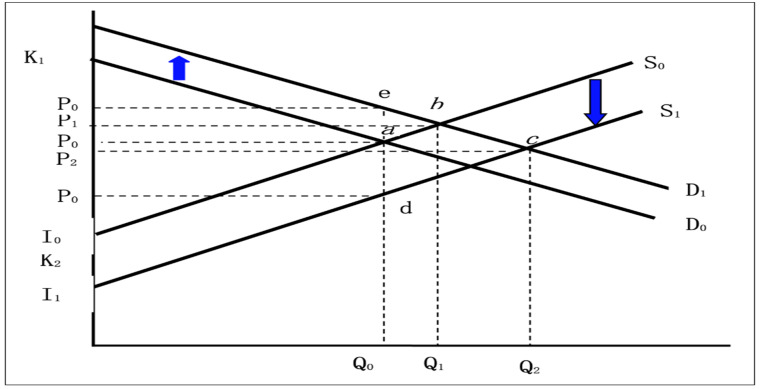
Supply and demand of kiwifruit commodity market brought by honeybee pollination technology.

**Table 1 ijerph-19-08305-t001:** Descriptive statistics of variables.

Variable Name	Variable Definition	Mean Value	Standard Deviation
**Explained variables**			
Honeybee pollination	Number of farmers who adopt honeybee pollination	0.16	0.36
Artificial pollination	Number of farmers who adopt artificial pollination	0.96	0.20
**Explanatory variables**			
**Farmer characteristics**			
Age	Age of the farmer	51.76	10.31
Education attainment	Junior high school and below = 1; high school or technical secondary school = 2; college degree or above = 3	1.40	0.64
Political affiliation	Party member = 1; non-party member = 0	0.21	0.41
Family characteristics			
Proportion of kiwi income	The proportion of kiwi planting in family income	47.16	31.65
Number of the labor force	The amount of labor engaged in Kiwi planting	2.26	1.78
**Management characteristics**			
Planting scale	Kiwi planting area (mu)	123.38	322.31
Planting experiences	Actual years of kiwi planting	8.72	5.31
† Product certification	Number of product certification	0.91	1.16
‡ Certificates of growers	Number of growers’ agricultural certificates	0.39	0.69
Access to purchase pollinating swarms	Yes = 1; no = 0	0.12	0.33
**Other related control variables**			
Cooperative membership	Yes = 1; no = 0	0.72	0.45
Distance to county	Distance (km)	21.04	12.73
Belief of the better quality of honeybee pollination	Better in honeybee pollination = 1; no better in honeybee pollination = 0;	0.63	0.48
Belief of the better quality of artificial pollination	Better in artificial pollination = 1; No better in artificial pollination = 0;	0.19	0.40

† Product certifications include pollution-free certification, green certification, organic certification, geographical indication certification, and special high-quality agricultural products certification. ‡ Farmer certificates include a certificate of the new farmer profession, a green certificate, and a certificate of an agricultural professional manager.

**Table 2 ijerph-19-08305-t002:** Comparison of the production costs and production efficiencies of both pollination technologies.

Item	Unit	Honeybee Pollination	Artificial Pollination	Net Gain	Change (%)
Fertilizer	yuan/mu	1672.67	1639.88	32.79	2.00
Pest control and pesticide	yuan/mu	237.66	208.27	29.39	14.11
Labor cost	yuan/mu	924.41	1154.27	−229.85	−19.91
Other inputs †	yuan/mu	1019.94	1242.17	−222.23	−17.89
Input total	yuan/mu	3854.68	4244.58	−389.90	−9.19
Yield	kg/mu	1073.43	930.48	142.95	15.36
Product price	yuan/kg	12.84	10.98	1.86	16.97
Output	yuan/mu	13,787.34	10,217.43	3569.90	34.94

† Other inputs include pollen purchase, bee swarm purchase, and land rent.

**Table 3 ijerph-19-08305-t003:** Estimated results of the bivariate probit model.

Variable	Honeybee Pollination			Artificial Pollination		
	Coefficient	Standard Error	Z Score	Coefficient	Standard Error	Z Score
**Farmer Characteristics**						
Age	**0.378 ****	0.169	2.24	−0.015	0.239	−0.63
Education level	0.386	0.340	1.13	0.121	0.326	0.37
Political affiliation	**0.818 ***	0.431	1.90	−0.425	0.509	−0.83
**Family Characteristics**						
Proportion of kiwi income	−0.0043	0.005	−0.63	−0.001	0.006	−0.12
Number of labor force	**−0.366 ****	0.177	−2.07	−0.044	0.168	−0.26
**Management Characteristics**						
Planting scale	**0.001 ***	0.001	1.64	0.001	0.000	0.03
Planting experiences	**−0.291 *****	0.084	−3.46	**0.248 ****	0.107	2.31
Product certification	−0.001	0.228	−0.01	**−0.452 ****	0.183	−2.46
Certificates of growers	**0.606 ***	0.321	1.89	−0.284	0.315	−0.90
Farmers’ access to purchase pollinating swarms	**5.142 *****	0.866	5.94	—	—	—
**Other related control variables**						
Cooperative membership	−0.547	0.461	−1.19	0.123	0.480	0.26
Distance to county	**−0.051 ****	0.023	−2.20	**0.154 *****	0.038	4.06
belief of the better quality of honeybee pollination	0.222	0.565	0.39	-0.099	0.617	−0.16
belief of the better quality of artificial pollination	**−1.346 ****	0.614	−2.19	**1.254 ***	0.733	1.71
**Constant term**	−0.469	1.258	−0.37	−0.765	1.653	−0.46
Log likelihood	=−14.826					
Observation	=185					

Note: ***, **, and * are, respectively, notable at 1%, 5% and 10%.

**Table 4 ijerph-19-08305-t004:** Economic benefits of honeybee pollination under two scenarios (ten thousand yuan).

Year	Scenario 1 (Adoption Rate = 50%)			Scenario 2 (Adoption Rate = 30%)		
	Producer Surplus	Consumer Surplus	Total Surplus	Producer Surplus	Consumer Surplus	Total Surplus
2016	26,552.41	39,333.64	65,886.05	26,552.41	39,333.64	65,886.05
2017	25,754.14	40,647.27	66,401.41	26,350.10	39,666.56	66,016.66
2018	23,373.40	44,565.00	67,938.40	25,289.04	41,412.64	66,701.68
2019	20,303.18	49,617.29	69,920.47	24,092.16	43,382.21	67,474.37
2020	18,536.09	52,525.16	71,061.24	23,274.28	44,728.11	68,002.39
2021	19,380.14	51,136.21	70,516.35	23,215.16	44,825.39	68,040.55
2022	21,732.01	47,266.03	68,998.04	23,820.06	43,829.98	67,650.04
2023	23,976.06	43,573.26	67,549.32	24,699.89	42,382.14	67,082.03
2024	25,502.89	41,060.73	66,563.62	25,528.84	41,018.02	66,546.86
2025	26,390.77	39,599.63	65,990.40	26,169.50	39,963.76	66,133.26
2026	26,453.31	39,496.72	65,950.03	25,670.30	40,785.24	66,455.54
Total	257,954.39	488,820.94	746,775.33	274,661.73	461,327.69	735,989.42

## Data Availability

The data that support our research findings are available from the corresponding author on request.

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
