# Peer review of "Does Adoption of Honeybee Pollination Promote the Economic Value of Kiwifruit Farmers? Evidence from China"

_ijerph, 2022, doi:10.3390/ijerph19148305_

Round 1

Reviewer 1 Report

Dear Authors,

I think some changes need to be made to the text to improve the article. I attached my comments in a file.

Sincerely yours,

Author Response

We are very grateful for your valuable time, constructive comments, and suggestions to help improve the quality of the manuscript.

Reviewer 2 Report

Good work needs minor revision.

Dear Dr.

Editor of Int. J. Environ. Res. Public Health

 Thank you very much for choosing me for reviewing to your esteemed Journal (Int. J. Environ. Res. Public Health). Please find the comments.

Abstract: Should be include background, objective, materials and methods, results, and conclusion. Abrief conclusion should answer the question in the title.

Introduction

·       The aim is not clear.

Discussion

·       discussion need more improvements.

Conclusion

·       There is too long and not clear.

·       Should answer the question in the title. 

Author Response

(The authors gave the same response as above.)

Reviewer 3 Report

Manuscript ijerph-1801481: Does Adoption of Honeybee Pollination Promote the Economic Value of Kiwifruit Farmers? Evidence from China.

The authors results showed that; (1) growers with higher social capital, proxied by political affiliation, are more aware of quality-oriented products, and older growers tend to choose less labor-intensive pollination technology; (2) with the increase in labor costs, more kiwifruit growers would choose honeybee pollination, and the more educated growers, measured by the number of training certificates, are more likely to adopt honeybee pollination; (3) lack of awareness and access to commercial pollinating swarms hinders the adoption of honeybee pollination; and (4) besides the economic benefit to producers, honey pollination also brings even larger consumer surplus. This study suggests some policy recommendations for promoting bee pollution in China

The data analysis methods are correct.

The English of the text is well written and well readable.

The uniqueness of the text is more than 90% by AntiPlagiarism.NET.

There are some comments and questions:

After the sentence - Honeybee pollination plays a significant role in maintaining the balance and biodiversity of sustainable agricultural system, rural development and ecosystems. - add citation (Ilyasov et al., 2020) and add to the references - Ilyasov, R.A.; Lee, M.-L.; Yunusbaev, U.B.; Nikolenko, A.G.; Kwon, H.-W. Estimation of C-derived introgression into A. m. mellifera colonies in the Russian Urals using microsatellite genotyping. Genes and Genomics 2020, 42, 987–996. doi: 10.1007/s13258-020-00966-0

There is no information on beekeeping status in the Sichuan province of China.

How many honey bee colonies, apiaries and beekeepers in the Sichuan province of China?

What species of honey bees used in beekeeping in the Sichuan province of China - Apis cerana and Apis mellifera, their proportion?

Why the bee pollination service is not developed  in the Sichuan province of China? Is there the environment pollution dangerous for surviving the pollinating insects in the Sichuan province of China?

Add information from answer the questions to the manuscript into discussion part.

Please improve the manuscript according to the above comments and answer the questions.

A minor revision is required.

Author Response

(The authors gave the same response as above.)

Reviewer 4 Report

The reported study is interesting, considering the importance of bees.

However, the paper requires moderate English changes. 

In particular, please avoid too many repetitions of the same word in the paragraph.

Line 169 - 211 is hard reading, so it should be rewritten.

The paper is sometimes confusing due to the mixing of data and methodology; it should be reorganized. Intending all report, data, results, discussion, and conclusions, some repetitions or parts are inserted in the wrong paragraph.

Many reported data are nonsense, with a standard deviation higher than the average.

Author Response

(The authors gave the same response as above.)
